# The Effect of Proximal Cleaning Devices on Periodontal Status in Korean Adults between 2016 and 2018

**DOI:** 10.3390/ijerph18042116

**Published:** 2021-02-22

**Authors:** Eun-Jeong Kim, Su-Jin Han

**Affiliations:** 1Department of Dental Hygiene, Gangdong University, Eumseong-gun 27600, Chungcheongbuk-do, Korea; kej1007@gangdong.ac.kr; 2Dental Research Institution, School of Dentistry, Seoul National University, Seoul 03080, Korea; 3Department of Dental Hygiene, College of Health Science, Gachon University, Incheon 21936, Korea

**Keywords:** dental devices, floss, epidemiology, interdental brush, periodontitis

## Abstract

Removal of the biofilm from the proximal space is essential for preventing periodontal disease. This study aimed to prove the association between the use of proximal cleaning devices, such as dental floss and interdental brushes, and periodontal health among nationally representative Korean adults. Data collected from the 7th National Health Nutrition Survey (KNHANES VII: 2016–2018) were used for this purpose. A total of 11,359 participants aged 19 years or older who participated in KNHANES were reviewed. The response variable was the prevalence of high CPI (CPI of 3–4), and the explanatory variables were dental floss and interdental brush. A multivariable logistic regression analysis was performed to adjust for potential confounding factors and to analyze the association between periodontal disease and proximal cleaning devices. It was found that 63.1% of the participants did not use proximal cleaning devices at all, 17.5% used dental floss alone, 11.9% used an interdental brush, and 7.5% used both. Subjects who used both dental floss and interdental brush had a high CPI rate nearly half that of all the models for those who did not. In particular, for those using dental floss, the aOR of high CPI was 0.681 in Model 1, 0.714 in Model 2, and 0.737 in Model 3. Dental hygiene products for cleaning the proximal space, such as dental floss, are essential for removing the dental biofilm as a basic tool along with toothbrushes. Teaching and explaining the need to use these devices well are important for oral health care and maintenance.

## 1. Introduction

Periodontal disease is one of the most common diseases and is a major cause of tooth loss [1]. Risk factors for developing periodontal disease include age, sex, stress, socioeconomic status, systemic disease, inappropriate host immune response, personal oral hygiene, and smoking [2]. Among them, the most important risk factors for periodontal disease are the composition of the subgingival bacteria. It is a chronic inflammatory plaque-related disease in which dental support structures are gradually destroyed [2,3]. Dental plaque is an etiological factor that causes the development of chronic periodontal disease, which, if untreated, may cause tooth loss [4,5]. Maintaining healthy oral hygiene by effectively removing the tooth surface bacterial plaque is the key approach for preventing and controlling periodontal disease [6]. Tooth brushing is the most basic method to prevent gingivitis and periodontal disease. However, by itself, it removes only 42% of the entire tooth surface bacterial biofilm [7,8]. Since the space between teeth is the main starting space for periodontal disease [9,10], removing the proximal biofilm is essential to prevent periodontal disease. It is recommended to clean the adjacent areas between teeth where a regular toothbrush cannot effectively reach [11].

Various devices can be used to clean the proximal spaces, but the appropriate ones should be selected according to an individual’s oral condition and their ability to use it. Representative methods that have received most attention for cleaning the proximal space are interdental brushes and dental floss. Recent studies have shown that daily dental flossing is mildly related to the low prevalence of gingivitis and periodontitis [12], and the American Dental Association (ADA) recommends flossing at least once daily for proximal cleaning [13]. There is also evidence that using an interdental brush along with a toothbrush is more effective against dental biofilms and gingivitis than using toothbrushes alone [14].

Unfortunately, despite the necessity of using these proximal cleaning devices, the proportion of people using them is still low. National evidence is also lacking to support the notion that self-care oral hygiene behavior is an essential tool for improving and maintaining good oral health. Moreover, there is weak and insufficient evidence of the effect of dental flossing and interdental brushing on the maintenance of dental health [14,15]. Since there is no strong, high-quality evidence to assess whether proximal cleaning behavior prevents periodontal disease [16], we attempted to confirm the present strength of association between the use of proximal cleaning devices, such as dental floss and interdental brushes, and periodontal health among nationally representative Korean adults. 

## 2. Materials and Methods

### 2.1. Study Design

This study used data collected from the 7th National Health Nutrition Survey (KNHANES VII: 2016–2018). The KNHANES VII was a cross-sectional survey conducted by the Korea Disease Control and Prevention Agency (KCDA) from 2016 to 2018, targeting civilians over the age of 1. The KNHANES sampling protocol was designed to include complex, stratified multistage and probability cluster surveys of representative samples of the uninstitutionalized civilian population of Korea. 

In this study, a multistage probability sampling unit was used, including geography, sex, and age determined according to the household register of the National Census Register, a domestic census for the last 5 years. Using census data, 192 primary sampling units (PSUs) were selected annually across Korea [17]. A total of 11,359 participants aged 19 years or older who participated in KNHANES were reviewed. Each subject completed a highly structured health survey and signed an informed consent form prior to participation in the study, which excluded subjects who did not answer one or more of the questions. Additionally, we analyzed all the variables considered in this study, except those that were missing. 

The KNHANES is a research conducted by the state directly for public welfare pursuant to the Bioethics and Safety Act and can be performed without deliberation by the Research Ethics Review Committee. Since 2018, research ethics have been reviewed in consideration of the collection of human-derived materials and provision of raw data to a third party, and the institutional review board of the KCDC approved the KNHANES (2018-01-03-P-A) [18].

### 2.2. Assessment of Periodontal Status

A trained dentist evaluated the subject’s periodontal status using the Community Periodontal Index of Treatment Needs (CPITN). According to the World Health Organization (WHO) guidelines, 10 teeth were selected and examined. The index teeth were two molars, the upper right, and the lower central front teeth. If the subject had no index teeth, the remaining teeth adjacent to it were examined. The score with the highest result score was recorded as the CPI score for each sextant. The periodontal health status of the participants with CPI was classified as 0 for healthy, 1 (gingivitis with bleeding on probing) to 2 (presence of calculus) for gingivitis and 3 (Periodontal Pocket Depth (PPD) ≥ 3.5 mm) to 4 (PPD ≥ 5.5 mm) for periodontitis. In this study, the periodontal health status was classified as high CPI if the CPI was 3–4. 

### 2.3. Assessment of the Use of Proximal Cleaning Devices

Proximal cleaning was measured based on the validated Korean version of the oral health questionnaire. The main question was “Please select all products that apply to your oral health except toothpaste and toothbrush.” Participants were asked about their usage of dental floss or interdental brushes and were accordingly categorized into four groups: none, dental floss, interdental brush, and both devices.

### 2.4. Assessment of Confounding Factors

The confounders of this study were the following major sociodemographic factors: sex, age, education, and household income. Age was classified into 7 groups: 19–24 years, 25–34 years, 35–44 years, 45–54 years, 55–64 years, 65–74 years, and ≥75 years. The level of education was classified into four groups based on the Korean education system: below primary school, middle school, high school, and college or higher education. The household income was classified into four groups: <25% (the lowest quartile group), 25–49%, 50–74%, and 75–100% (the highest quartile group).

The health behavior covariates included smoking, tooth-brushing frequency, dental clinic visits within a year, and proximal cleaning. Participants were categorized into three groups based on their smoking experience: “never smoked”, “past smoker,” and “current smoker.” The daily tooth-brushing frequency was categorized as once or less, twice, and three or more times. Dental clinic visits were classified as “yes” or “no,” based on the dental clinic visits in the past 1 year. The health status covariates included diabetes mellitus, hypercholesterolemia, hypertension, and obesity. With respect to diabetes mellitus, the participants were classified into three groups: normal, impaired fasting glucose (fasting blood glucose: 100–125.9 mg/dL), and diabetic (fasting blood glucose ≥ 126 mg/dL or on medication or insulin). Hypercholesterolemia was classified into two groups: normal and abnormal (total cholesterol ≥ 240 mg/dL or on medication). Hypertension was classified into three groups: normal, pre-hypertensive (systolic blood pressure 130–139.9 mmHg, or diastolic blood pressure 85–89.9 mmHg), and hypertension (systolic blood pressure ≥ 140 mmHg or diastolic blood pressure ≥ 90 mmHg or on medication). Body Mass Index (BMI) was classified into three different groups, according to the WHO Asia-Pacific Guidelines (World Health Organization. Regional Office for the Western 2000): underweight (<18.5 kg/m^2^), normal (18.5–24.9 kg/m^2^), and obese (≥25.0 kg/m^2^).

### 2.5. Statistical Analysis

The response variable was the prevalence of high CPI (CPI 3–4) and the explanatory variable was proximal cleaning (use of dental floss and/or interdental brush). The statistics of the subject characteristics was analyzed by frequency and percentage. Weights were applied to estimate the weight proportions, using the chi-square test of complex sample analysis for determining statistically significant differences in the characteristics of subjects, depending upon the use of proximal cleaning devices in terms of socio-economic, behavioral, general, and oral health-related factors.

A multivariable logistic regression analysis was performed to adjust for potential confounders and to analyze the association between periodontal disease and proximal cleaning devices. Regression model 1 was adjusted for age, sex, household income, and education level. Personal health practice variables, such as smoking, tooth-brushing, and dental clinic visits, were added to regression model 2. Systematic medical factor variables, such as diabetes mellitus, hypercholesterolemia, hypertension, and obesity were added to regression model 3. Data analysis was conducted using IBM SPSS ver. 25.0 (IBM Co., Armonk, NY, USA) and complex sample analysis was conducted with stratification variables, random, cluster, and weights for all analyses. *p* < 0.05 was considered to be statistically significant.

## 3. Results

### 3.1. Characterization of the Study Population by High CPI

A total of 11,359 Korean adults (4984 men and 6375 women) were included in this study, of which, 30.4% had high CPI (CPI 3–4) (Table 1). Of those with high CPI (CPI 3–4), 74.4% did not use proximal cleaning devices, 11.2% used only dental floss, 10.8% used only the interdental brush, and 3.6% used both. Furthermore, 12.9% of the patients brushed their teeth once or less per day. Those with low CPI also had a higher rate of visiting to the dentist regularly.

### 3.2. Clinical Characteristics of those Using Proximal Cleaning Devices

Of the total subjects, 63.1% did not use proximal cleaning devices, 17.5% used dental floss alone, 11.9% used interdental brush, and 7.5% used both. The maximum use of dental floss was made by subjects aged 35–44, while those aged 45–54 were the largest users of interdental brushes. The better the educational background and higher the household income, the higher the rate of using proximal cleaning devices. Furthermore, 83.1% of those who brushed their teeth once or less a day did not use proximal cleaning devices, while those who brushed their teeth more than three times a day showed a high rate of using proximal cleaning devices. Those who used all of the proximal cleaning devices were found to have a more positive tooth-brushing frequency, non-smoking, and dental visit practices than those who did not use them. In addition, the prevalence of diabetes, hypercholesterolemia, hypertension, and high CPI was significantly lower (Table 2).

### 3.3. Association between Proximal Cleaning Devices and High CPI (CPI 3–4)

Table 3 shows the outcomes of the logistic regression analysis using proximal cleaning devices and high CPI (CPI 3–4). Subjects who used both dental floss and interdental brushes had a high CPI rate nearly half that of all models for those who did not use them (aOR, 0.581; 95% CI, 0.456–0.740 for Model 1; aOR, 0.599; 95% CI, 0.469–0.765 for Model 2; aOR, 0.591; 95% CI, 0.462–0.756 for Model 3). For those using dental floss, the aOR of high CPI was 0.681 (95% CI, 0.581–0.799) in Model 1, 0.712 (95% CI, 0.605–0.838) in Model 2, and 0.734 (95% CI, 0.623–0.865) in Model 3. Conversely, the interdental brush showed no significant aOR.

## 4. Discussion

Several studies have shown that the use of oral hygiene products is related to periodontal health [8,14,15], but accurate epidemiological studies on a large population are still lacking. The data from this study on Korean national adults present clear evidence that the use of dental floss and a combination of dental floss and interdental brush are associated with a low prevalence of periodontal diseases. These can, therefore, be practical tools for promoting oral health. 

The results of this study showed that the frequency of tooth-brushing, visits to the dentist, and the use of proximal cleaning devices tended to be similar to the prevalence of high CPI, and the frequency of tooth-brushing and the use of interdental brushes have been shown to relieve high CPI, as established by previous studies [19]. This study showed that only 17.5% of subjects used dental floss and 11.9% used interdental brushes, while only 7.5% used both, and 63.1% did not use anything at all. This result was similar to that of the Delta national poll of 1003 adults, in which 20% of Americans were found to not use dental floss [20]. Another study reported that 32% of adults do not use dental floss [21]. In previous studies, the use of interdental brushes in relation to plaque scores, bleeding scores, and pocket depth has shown positive effects [22,23,24]. However, in some Korean studies, there was no significant difference between those who used interdental brushes and those who did not [25], similar to the findings of this study. However, the current study showed a positive effect of dental floss on reducing the prevalence of high CPI. 

There is a growing debate about the efficacy of dental floss and its role in periodontal disease [15,26,27], but it is a recommended part of daily dental hygiene. According to the American Dental Association, dental floss is most widely used for proximal cleaning [11,15,26,28]. The report estimates that 80% of the proximal biofilm can be removed with dental floss, significantly reducing the prevalence of periodontal disease [13,29]. The current study is the first to report the nationwide prevalence of high CPI in proximal cleaning devices for periodontal disease in Korean adults. The results of this study found that dental floss alone, and along with interdental brushes, significantly reduced high CPI. In a previous study on the use of oral hygiene products in subjects with implants, individuals over 60 years of age used interdental brushes more than dental floss, and the results were similar to those of this study’s subjects over 75 years of age using an interdental brush more than dental floss. Older people have difficulty using large and small muscles, making it more convenient to use an interdental brush rather than dental floss, which can be a barrier to older people with physical limitations, affecting oral hygiene behavior and dental care. Not only the elderly but also the general public accepts oral hygiene as a means of removing food debris, but they often do not understand the act of removing tooth biofilm [30].

Dental caries and periodontal disease can occur as biofilms build up on the teeth and gingiva. Of the various anatomical structures of teeth, cleaning the gingival sulcus, where bacteria are present and plaque accumulates most, is the most important [31]. Dental floss can effectively reach all areas, except deep periodontal pockets and furcation areas. Unfortunately, this study was unable to analyze whether dental floss was used correctly. However, it is believed that teaching and practicing the correct use of dental floss can help maintain and improve oral health much more effectively. Therefore, not only should the point and method of using proximal cleaning devices be emphasized but the effects and reasons to teach patients effectively should also be recognized. Since is very difficult to change one’s habits and behavior, rather than expecting immediate changes in their actions, it is important to motivate the subjects significantly, so that they can gradually gain an understanding and cooperate.

However, this study is not without limitations. First, this study has been evaluated only for its association with cross-sectional studies. A prospective study is needed to determine whether the use of proximal cleaning devices can reduce periodontal disease. Second, it is difficult to objectively analyze the response to the use of proximal cleaning devices using a self-questionnaire tool. In addition, it is difficult to check whether proximal cleaning devices were used in the correct method. In future research, studies and analyses that complement these points should be conducted. Finally, the CPI was used to assess the periodontal status; CPI checks the condition of periodontitis on a representative tooth, which may include pseudo-pockets. In this case, the prevalence of periodontitis may have been overestimated or underestimated. In addition, also, according to AAP workshop 2017 [2], classification of periodontal disease and conditions, the main clinical criterion for periodontitis is the clinical attachment level/loss. Therefore, it is necessary to consider that there is a bias in our results regarding the diagnosis of periodontitis. Notwithstanding these limitations, the findings of this study support the hypothesis that proximal cleaning devices, such as dental floss and interdental brushes, are associated with a lower prevalence of high CPI.

## 5. Conclusions

The results of this study show that along with toothbrushes, dental hygiene products for cleaning the proximal spaces, such as dental floss, are essential as a basic tool for removing the dental biofilm. Teaching and explaining the need to use these devices well is important for better oral health care and maintenance.

## Figures and Tables

**Table 1 ijerph-18-02116-t001:** Characteristics of the study population with periodontal status.

		Total		Periodontal Status				*p*
				High CPI (CPI 3–4)		Low CPI (CPI 0–2)		
All		11,359		3529	(30.4)	7830	(69.6)	
Sex	Male	4984	(42.3)	1884	(51.9)	3100	(38.1)	<0.001
	Female	6375	(57.7)	1645	(48.1)	4730	(61.9)	
Age, year	19–24	799	(7.5)	25	(0.9)	774	(10.4)	<0.001
	25–34	1406	(12.5)	108	(3.0)	1298	(16.6)	
	35–44	2197	(18.6)	437	(11.5)	1760	(21.6)	
	45–54	2201	(19.7)	745	(21.5)	1456	(18.9)	
	55–64	2215	(19.9)	988	(28.6)	1227	(16.2)	
	65–74	1626	(14.1)	778	(22.0)	848	(10.7)	
	≥75	915	(7.7)	448	(12.6)	467	(5.6)	
Education	≤Elementary school	2083	(17.6)	1042	(28.9)	1041	(12.6)	<0.001
	Middle school	1094	(9.8)	481	(14.4)	613	(7.8)	
	High school	3749	(33.7)	1075	(30.8)	2674	(35.0)	
	≥University or college	4433	(38.9)	931	(25.9)	3502	(44.6)	
Household income	Lowest	1982	(17.1)	861	(24.2)	1121	(14.0)	<0.001
	Middle low	2745	(23.8)	954	(26.7)	1791	(22.6)	
	Middle high	3240	(28.4)	900	(25.5)	2340	(29.7)	
	Highest	3392	(30.7)	814	(23.5)	2578	(33.8)	
Smoking	Non-smoker	6962	(62.5)	1779	(51.4)	5183	(67.3)	<0.001
	Ex-smoker	2371	(20.2)	887	(24.8)	1484	(18.3)	
	Current smoker	2026	(17.2)	863	(23.8)	1163	(14.4)	
Using proximal cleaning devices	Dental floss	1966	(17.5)	386	(11.2)	1580	(20.2)	<0.001
	Interdental brush	1359	(11.9)	400	(10.8)	959	(12.4)	
	Both devices	831	(7.5)	133	(3.6)	698	(9.2)	
	None	7203	(63.1)	2610	(74.4)	4593	(58.2)	
Tooth-brushing frequency	≤1/day	1073	(8.9)	470	(12.9)	603	(7.1)	<0.001
	2/day	4372	(38.4)	1488	(42.4)	2884	(36.6)	
	≥3/day	5914	(52.8)	1571	(44.7)	4343	(56.3)	
Dental clinic visits within a year	No	4716	(41.6)	1500	(42.8)	3216	(41.1)	0.180
	Yes	6643	(58.4)	2029	(57.2)	4614	(58.9)	
Diabetes	Normal	7267	(64.9)	1738	(51.0)	5529	(71.0)	<0.001
	Impaired fasting glucose	2711	(23.4)	1070	(29.0)	1641	(21.0)	
	Diabetes	1381	(11.6)	721	(20.0)	660	(8.0)	
Hypercholesterolemia	Normal	8833	(77.6)	2572	(72.5)	6261	(79.8)	<0.001
	Abnormal	2526	(22.4)	957	(27.5)	1569	(20.2)	
Hypertension	Normal	5108	(45.6)	1063	(30.2)	4045	(52.3)	<0.001
	Prehypertension	2748	(24.3)	902	(25.8)	1846	(23.7)	
	Hypertension	3503	(30.1)	1564	(44.0)	1939	(24.0)	
BMI	Underweight	413	(3.6)	77	(2.0)	336	(4.4)	<0.001
	Normal	7019	(62.8)	1992	(57.8)	5027	(65.0)	
	Obese	3927	(33.6)	1460	(40.2)	2467	(30.7)	

*p*-values were obtained through complex sample cross-tabs.

**Table 2 ijerph-18-02116-t002:** Characteristics of the study population using dental floss and/or interdental brush.

		Using Dental Floss and/or Interdental Brush								*p*
		None		Dental Floss		Interdental Brush		Both Devices		
All		7203	(63.1)	1966	(17.5)	1359	(11.9)	831	(7.5)	
Sex	Male	3466	(46.5)	672	(32.9)	574	(41.5)	272	(30.4)	<0.001
	Female	3737	(53.5)	1294	(67.1)	785	(58.5)	559	(69.6)	
Age, year	19–24	534	(8.0)	114	(5.9)	108	(8.7)	43	(5.1)	<0.001
	25–34	637	(9.0)	331	(16.8)	230	(17.3)	208	(24.4)	
	35–44	1013	(13.3)	586	(29.0)	330	(22.4)	268	(32.2)	
	45–54	1341	(18.8)	412	(21.7)	268	(19.7)	180	(22.6)	
	55–64	1575	(22.2)	343	(17.9)	212	(16.7)	85	(10.5)	
	65–74	1309	(18.0)	141	(7.0)	140	(10.0)	36	(4.2)	
	≥75	794	(10.6)	39	(1.7)	71	(5.3)	11	(1.2)	
Education	≤Elementary school	1799	(24.1)	103	(5.0)	158	(11.1)	23	(2.6)	<0.001
	Middle school	856	(12.3)	103	(5.2)	106	(7.3)	29	(3.6)	
	High school	2390	(34.0)	614	(31.7)	497	(37.7)	248	(29.3)	
	≥University or college	2158	(29.6)	1146	(58.1)	598	(43.9)	531	(64.6)	
Household income	Lowest	1583	(21.4)	168	(8.9)	185	(13.5)	46	(5.5)	<0.001
	Middle low	1820	(24.9)	431	(20.8)	310	(23.6)	184	(22.8)	
	Middle high	1954	(27.2)	628	(32.0)	392	(27.7)	266	(31.2)	
	Highest	1846	(26.5)	739	(38.3)	472	(35.2)	335	(40.4)	
Smoking	Non-smoker	4218	(60.0)	1361	(69.9)	833	(61.7)	550	(67.4)	<0.001
	Ex-smoker	1607	(21.6)	355	(17.5)	252	(18.6)	157	(17.8)	
	Current smoker	1378	(18.4)	250	(12.6)	274	(19.7)	124	(14.8)	
Tooth-brushing frequency	≤1/day	892	(11.7)	78	(4.0)	81	(5.3)	22	(2.4)	<0.001
	2/day	3039	(42.0)	647	(32.7)	457	(34.4)	229	(27.2)	
	≥3/day	3272	(46.3)	1241	(63.4)	821	(60.3)	580	(70.4)	
Dental clinic visits within a year	No	3233	(45.4)	675	(34.1)	562	(40.2)	246	(29.3)	<0.001
	Yes	3970	(54.6)	1291	(65.9)	797	(59.8)	585	(70.7)	
Diabetes	Normal	4307	(60.8)	1458	(74.8)	875	(64.8)	627	(76.6)	<0.001
	Impaired fasting glucose	1851	(25.5)	387	(19.2)	321	(22.9)	152	(17.2)	
	Diabetes	1045	(13.7)	121	(6.1)	163	(12.3)	52	(6.2)	
Hypercholesterolemia	Normal	5492	(76.0)	1598	(81.0)	1056	(77.5)	687	(83.1)	<0.001
	Abnormal	1711	(24.0)	368	(19.0)	303	(22.5)	144	(16.9)	
Hypertension	Normal	2837	(39.8)	1149	(59.3)	627	(47.0)	495	(59.8)	<0.001
	Prehypertension	1761	(24.7)	432	(22.0)	360	(26.6)	195	(23.5)	
	Hypertension	2605	(35.5)	385	(18.8)	372	(26.4)	141	(16.7)	
BMI	Underweight	211	(2.8)	109	(5.8)	53	(4.2)	40	(4.8)	<0.001
	Normal	4398	(62.1)	1329	(68.2)	777	(57.9)	515	(63.5)	
	Obese	2594	(35.1)	528	(26.0)	529	(38.0)	276	(31.8)	
Periodontal disease	Low CPI (CPI 0–2)	4593	(64.2)	1580	(80.5)	959	(72.4)	698	(85.4)	<0.001
	High CPI (CPI 3–4)	2610	(35.8)	386	(19.5)	400	(27.6)	133	(14.6)	

*p*-values were obtained through complex samples cross-tabs.

**Table 3 ijerph-18-02116-t003:** Multivariable association between using dental floss and/or interdental brush and high CPI (CPI 3-4).

		Model 1	*p*	Model 2	*p*	Model 3	*p*
		OR (95% CI)		OR (95% CI)		OR (95% CI)	
Using dental floss and/or interdental brush	Dental floss	0.681 0.581–0.799	<0.001	0.712 0.605–0.838	<0.001	0.734 0.623–0.865	<0.001
	Interdental brush	0.983 0.834–1.157	0.832	0.990 0.838–1.170	0.908	0.963 0.816–1.138	0.660
	Dental floss and interdental brush	0.581 0.456–0.740	<0.001	0.599 0.469–0.765	<0.001	0.591 0.462–0.756	<0.001
	None	Reference		Reference		Reference	

Response variable: high CPI. Explanatory variable: dental floss and/or interdental brush use. Model 1 was adjusted to the socioeconomic status variables (sex, age, education levels, and household income). Model 2 was additionally adjusted to personal health practice variables (smoking, tooth-brushing, and dental clinic visit). Model 3 was additionally adjusted to systematic medical factor variables (diabetes mellitus, hypercholesterolemia, hypertension, and obesity). CPI; Community Periodontal Index.

## Data Availability

Publicly available datasets were analyzed in this study. These data can be found here: [https://knhanes.cdc.go.kr/knhanes/sub03/sub03_02_05.do] (accessed on 10 December 2020).

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
