# Peer review of "The Effect of Proximal Cleaning Devices on Periodontal Status in Korean Adults between 2016 and 2018"

_ijerph, 2021, doi:10.3390/ijerph18042116_

Round 1

Reviewer 1 Report

The article is informative.

(1). However clarification regarding few observations would help to improve the article, such as more comprehensive explanation of risk factors of periodontal diseases in the introduction.

(2). The material and methods briefly explains the chosen design of carried out study. Results are clearly explained.

(3). Although, the quality of toothbrushing or presence of dental plaque is not analysed.

(4). Furthermore, marginal periodontitis is less prevalent in young population, while gingivitis (CPI=1, 2) is more widespread among people.

(5). Finally, a long-lasting gingivitis can cause periodontitis.

(6). Discussion- comparison with similar studies is too less.

Author Response

Thanks for reviewing my research. Below are the answers to your review. I tried to do my best. Thank you very much.

Our revised paper have been checked by a native English speaker (American Journal Experts).

 Comments and Suggestions for Authors

The article is informative.

(1). However clarification regarding few observations would help to improve the article, such as more comprehensive explanation of risk factors of periodontal diseases in the introduction.

Reply: We added the more comprehensive explanation of risk factors of periodontal disease mentioned by reviewers in the introduction.

(2). The material and methods briefly explains the chosen design of carried out study. Results are clearly explained.

Reply: This study was designed by reflecting the design of the number of preceding studies analyzed with data from the Korean National Health Nutrition Survey. What part of the method did you think was concise? We think the authors have filled in enough, but if you tell us what the reviewer thinks you are lacking in the method, we will fill in additional details.

(3). Although, the quality of toothbrushing or presence of dental plaque is not analysed.

Reply: Unfortunately, no survey or clinical data on the quality of tooth brushing is included. Also, there are no levels of the dental plaque. However, if the adjacent surface is less cleaned or not cleaned, the presence of a dental plaque naturally increases the probability of developing periodontitis. Therefore, the authors chose periodontitis as the resultant indicator. We would very appreciate it if reviewer consider the authors’ opinions. Thank you for your advice.

(4). Furthermore, marginal periodontitis is less prevalent in young population, while gingivitis (CPI=1, 2) is more widespread among people.

Reply: This study analyzed the results of measuring periodontitis with CPITN as the standard classified by WHO as an outcome variable. Therefore, as described in the method section of the manuscript, the periodontal health status of the participants with CPI was classified as 0 for healthy, 1 (gingivitis with bleeding on probing) to 2 (presence of calculus) for gingivitis, and 3 (Periodontal Pocket Depth (PPD) ≥3.5 mm) to 4 (PPD ≥5.5 mm) for periodontitis. In this study, the periodontal health status was classified as periodontitis if the CPI was 3–4 or higher. The marginal periodontitis mentioned by reviewers were not separately classified and analyzed. More details are provided in the text.

(5). Finally, a long-lasting gingivitis can cause periodontitis.

Reply: We agree with the reviewer. The contents of the manuscripts also contain the above.

(6). Discussion- comparison with similar studies is too less.

Reply: As reviewer said, author added more recent articles in the discussion.

Reviewer 2 Report

Hypothesis. Page 2 line 55, instead of prove the association, authors should consider to use ‘present the strength of association’ methodology used for the study is to detect associations.

All comparisons present significant differences (p values) in table 2 but it is not clear in the tables which variables lead to significant differences.

Table 2 instead of Normal, use No Periodontitis or use CPI 0-2 and CPI 3-4

Table 3 how do authors explain why patients who use of dental floss and/or interdental brush don’t present as protective effects against periodontitis when patients who use dental floss only and dental floss and inter proximal brush has significantly less odd ratios for periodontal disease?

The discussion could be improved with more current literature comparisons.

Authors should include more recent articles on inter proximal cleaning aids.

 Home use of inter dental cleaning devices, in addition to tooth brushing, for preventing and controlling periodontal diseases and dental caries.

Worthington HV, MacDonald L, Poklepovic Pericic T, Sambunjak D, Johnson TM, Imai P, Clarkson JE.Cochrane Database Syst Rev. 2019 Apr 10;4(4):CD012018. doi: 10.1002/14651858.CD012018.pub2.PMID: 30968949 Free PMC article.

Efficacy of inter-dental mechanical plaque control in managing gingivitis--a meta-review. Sälzer S, Slot DE, Van der Weijden FA, Dörfer CE.J Clin Periodontol. 2015 Apr;42 Suppl 16:S92-105. doi: 10.1111/jcpe.12363.PMID: 25581718 Review.

Author Response

Thanks for reviewing my research. Below are the answers to your review. I tried to do my best. Thank you very much.

Our revised paper have been checked by a native English speaker (American Journal Experts).

Hypothesis. Page 2 line 55, instead of prove the association, authors should consider to use ‘present the strength of association’ methodology used for the study is to detect associations.

Reply: The authors agreed with the reviewer’s comments and revised the hypothesis. Thank you for pointing it out.

All comparisons present significant differences (p values) in table 2 but it is not clear in the tables which variables lead to significant differences.

Reply: We added to the manuscript a description of the difference between health behaviors and health status according to the use of proximal cleaning devices.

Table 2 instead of Normal, use No Periodontitis or use CPI 0-2 and CPI 3-4

Reply: We revised all the points you pointed out.

Table 3 how do authors explain why patients who use of dental floss and/or interdental brush don’t present as protective effects against periodontitis when patients who use dental floss only and dental floss and inter proximal brush has significantly less odd ratios for periodontal disease?

Reply: The results showed that the odds ratio of developing periodontitis when used proximal cleaning devices together was lower. Also, we think there will be a content of the results from this part because it is not easy to use the proximal cleaning devices. In this study, we thought there would be a limitation in this part because there was no question or clinical check about whether or not the proximal cleaning devices was used correctly. We added that part to the limitation in discussion part.

The discussion could be improved with more current literature comparisons.Home use of inter dental cleaning devices, in addition to tooth brushing, for preventing and controlling periodontal diseases and dental caries.Efficacy of inter-dental mechanical plaque control in managing gingivitis--a meta-review. Sälzer S, Slot DE, Van der Weijden FA, Dörfer CE.J Clin Periodontol. 2015 Apr;42 Suppl 16:S92-105. doi: 10.1111/jcpe.12363.PMID: 25581718 Review.

Worthington HV, MacDonald L, Poklepovic Pericic T, Sambunjak D, Johnson TM, Imai P, Clarkson JE.Cochrane Database Syst Rev. 2019 Apr 10;4(4):CD012018. doi: 10.1002/14651858.CD012018.pub2.PMID: 30968949 Free PMC article.

Authors should include more recent articles on inter proximal cleaning aids.

Reply: We added the recent articles reviewer mentioned in the discussion.

Reviewer 3 Report

Very well conducted research with large sample size and very well written manuscript. A follow up research study with prospective study design to reduce the bias associated with self reported data will be very helpful.

Author Response

Thanks for reviewing my research. Below are the answers to your review. I tried to do my best. Thank you very much.

Our revised paper have been checked by a native English speaker (American Journal Experts).

Round 2

Reviewer 2 Report

The hypothesis sentence should not include ‘prove’

According to AAP workshop 2017, Classification of Periodontal Diseases and Conditions, the main clinical criteria for periodontitis is the clinical attachment level/loss. CPI classification of periodontitis is not supporting new periodontal disease classification criteria which indicates that authors classification of disease might be incorrect.

Results and discussion sections need significant improvement to support authors’ decision of the results.

Author Response

Reviewer 2.

Thanks for reviewing my research. Below are the answers to your review. I tried to do my best. Thank you very much.

Our revised paper have been checked by a native English speaker (American Journal Experts).

Comments and Suggestions for Authors

The hypothesis sentence should not include ‘prove’

  • The authors agreed with the reviewer’s comments and revised. Thank you for pointing it out.

According to AAP workshop 2017, Classification of Periodontal Diseases and Conditions, the main clinical criteria for periodontitis is the clinical attachment level/loss. CPI classification of periodontitis is not supporting new periodontal disease classification criteria which indicates that authors classification of disease might be incorrect.

  • We agree to your opinion. It would be much better if periodontitis could be defined by CDC-AAP definition or measurement of CAL and PD. However, the CPI proposed by WHO has been widely and successfully used in the epidemiologic surveys and analyses for the association between periodontitis and systemic conditions(Lee et al., 2013, Ahn et al., 2015, Lee et al., 2017). Hence, we author presented the limitations of CPI in the discussion section, but supplemented further. We would appreciate it if you understand the intention of the author.
  • Ahn, Y. B., Shin, M. S., Byun, J. S. & Kim, H. D. (2015) The association of hypertension with periodontitis is highlighted in female adults: results from the Fourth Korea National Health and Nutrition Examination Survey. J Clin Periodontol 42, 998-1005. doi:10.1111/jcpe.12471.
  • Lee, J. B., Yi, H. Y. & Bae, K. H. (2013) The association between periodontitis and dyslipidemia based on the Fourth Korea National Health and Nutrition Examination Survey. J Clin Periodontol 40, 437-442. doi:10.1111/jcpe.12095.
  • Lee, S. W., Lim, H. J. & Lee, E. (2017) Association Between Asthma and Periodontitis: Results From the Korean National Health and Nutrition Examination Survey. J Periodontol 88, 575-581. doi:10.1902/jop.2017.160706.

Results and discussion sections need significant improvement to support authors’ decision of the results.

  • We added and revised in the results and discussion.
